# Smart Total Knee Replacement: Recognition of Activities of Daily Living Using Embedded IMU Sensors and a Novel AI Model in a Cadaveric Proof-of-Concept Study [note 1]

**DOI:** 10.3390/s25216657

**Published:** 2025-10-31

**Authors:** Lipalo Mokete, Alexander Conway, Emma Donnelly, Ryan Willing

**Affiliations:** 1Moriana Innovations, Johannesburg 2191, South Africa; alex@connectedknee.com; 2Division of Orthopaedic Surgery, University of the Witwatersrand Johannesburg, Johannesburg 2193, South Africa; 3School of Biomedical Engineering, Western University, London, ON N6A 3K7, Canada; edonne@uwo.ca (E.D.); rwilling@uwo.ca (R.W.); 4Bone and Joint Institute, Western University, London, ON N6G 2V4, Canada; 5Department of Mechanical and Materials Engineering, Western University, London, ON N6A 5B9, Canada

**Keywords:** smart implant, knee replacement, AI, sensors, activity of daily living

## Abstract

**Highlights:**

**What are the main findings?**

**What is the implication of the main finding?**

**Abstract:**

Total knee replacement (TKR) is a reliable treatment for end-stage degenerative conditions of the knee. Patient-reported outcome measures (PROMs) are central to assessing TKR outcomes, but they have limitations. Activities of daily living (ADLs) in the early post-operative period complement PROMs for holistic patient assessment. This study presents a method for capturing ADL parameters from data generated by inertial measurement unit (IMU) devices embedded in TKR prosthesis. A conventional posterior stabilized TKR was modified to create chambers in the femoral and tibial components. The prosthesis was implanted into a cadaver knee and movement was simulated using a hydraulic actuated knee simulator (AMTI, VIVO, MA, USA). A powered IMU device was placed in each of the chambers. The simulator was activated for various ADLs and the generated data was collected wirelessly. The pre-processed data was fed into a novel multimodal deep learning artificial intelligence model created to recognize specific ADL (trained on 70% of the data, with 30% reserved for validation and testing). The model achieved 95.68% overall accuracy, with 100% for sitting, standing, stance, and knee bending. Walking, stair navigation, and jogging showed F1 scores of 0.98, 0.92, 0.91, and 0.89, respectively. This technology enables seamless knee activity recognition and reporting with positive implications for patient-specific rehabilitation protocols.

## 1. Introduction

Human locomotion involves complex interactions between bones, joints, and soft tissues. Current gold standard movement analysis relies on specialized gait laboratories combining videography with limb and joint trackers [1]. However, these facilities are expensive to set up and access, require specialized expertise, and are created in controlled environments that may not reflect real-world conditions [2,3].

Inertial measurement units (IMUs), comprising accelerometers and gyroscopes, offer a promising solution for objective locomotion assessment in natural environments [2]. The devices have gained widespread adoption in wearable form, particularly in the health and fitness industry, due to their accessibility and cost-effectiveness [4]. IMU data interpretation enables an objective measurement of key mobility metrics, including step count, stride length, and gait patterns [5].

Osteoarthritis affects over 500 million people globally, with the knee being the most affected joint. As the disease progresses, it significantly impacts quality of life by limiting activities of daily living (ADLs) [6]. For end-stage disease, total knee replacement has become the standard treatment, with well over a million procedures performed annually worldwide [7]. Although TKR restores quality of life, the procedure is associated with significant healthcare costs. Current initiatives focus on cost containment while improving patient outcomes through personalized medicine approaches [8].

Inherent in the improvement of quality of life following total knee replacement (TKR) is the ability to execute ADLs with minimal impediment. While validated PROMs are the standard for evaluating TKR outcomes, they lack objectivity and suffer from ceiling and floor effects [9]. Several authors have found a lack of correlation between patient-reported outcome measures (PROMs) and objective activity measures in the early post-TKR period, and there are now calls for physical activity tests (ADLs) to be included in primary outcome measures [10,11,12]. However, continuous direct observation of ADLs during rehabilitation is impractical, and wearables are subject to compliance challenges [13].

Wearable devices have been used to establish normative values for daily functional recovery patterns following TKR based on total daily steps and cadence [14]. Wearables have also demonstrated accurate knee angle measurements post-TKR across ADLs [15]. A recent study on an implanted IMU housed in a tibial extension of a knee replacement prosthesis showed objective increases in gait parameters in the first six weeks [16]. This IMU-generated activity data from implanted devices has been used to create recovery curves based on quantitative metrics like step counts [17]. Published TKR research involving the use of IMUs has focused on mobility metrics largely excluding the wider spectrum of ADLs [18]. Specific ADLs studied using a wearable activity monitor following TKR showed that the most frequently performed activity was standing, followed by level walking, sitting, stair walking, and lying down [19].

The growing ability to miniaturize not only IMU sensor devices but long-life implantable batteries has made the possibility of embedding such powered sensors within cavities in knee replacement prosthesis a reality [20]. The sensors are capable of wirelessly transmitting activity data of the knee replacement and they can be placed within the tibia, femur, or articulating spacer component of the knee prosthesis. We hypothesize that data generated from repetitions of specific ADLs can be used to build an artificial intelligence (AI) model that recognizes the pattern of generated data and is able to accurately predict the ADL. In this manuscript, we present ADL recognition based on data generated by multiple IMU devices embedded in a knee replacement in a simulated cadaveric study.

## 2. Materials and Methods

The experimental study was carried out at the Mechanical and Materials Engineering Department Laboratory, Western University, Canada. Institutional ethics approval was obtained before the commencement of the study (project ID 123533). An entire left lower limb was sourced from the cadaver of an 81-year-old man with no osteoarthritis of the knee. A CT scan of the limb was performed to confirm adequacy of the specimen, exclude pre-existing deformities and retained hardware, and for purposes of sizing of the knee for the appropriate TKR prosthesis. A size 7 posterior stabilized knee (5C, PS, Implantcast, Buxtehude, Germany) was modified by creating blind chambers in both the femoral and tibial components. The chamber on the lateral aspect of the femur was created by removing metal from the posterior condyle. Because there was not enough space to create a similar chamber in the tibia, a full circumference titanium alloy augment with a blind lateral chamber was created using rapid prototyping technology. The augment was created from 3-dimensional CAD drawings and manufactured using laser sintering technology for additive manufacturing on an EOSINT M280 3D printer (Centre for Rapid Prototyping and Manufacture, Bloemfontein, South Africa) and affixed securely to the underside of a size 7 tibial component using screws. The limb was mounted and prepared for surgery.

Surgery was performed by an experienced fellowship trained joint replacement surgeon (LM). A midline incision was made with a medial parapatella deep approach. The patella was subluxed laterally, and bony cuts were made based on intramedullary guidance on the femur and extramedullary guidance on the tibia using standard conventional jigs. Equal flexion and extension gaps were achieved for a size 7 implant and a 10 mm polyethylene component (Figure 1, Figure 2, Figure 3, Figure 4 and Figure 5). The femoral box cut was made, and additional bone was resected from the tibia to accommodate the augment. The resected bone surfaces were prepared for cementation, and the modified prosthesis was implanted using Simplex^®^ bone cement (Stryker corporation, Kalamazoo, MI, USA). The prosthesis was stable with a full range of motion. The limb was filleted to expose the bone shaft at mid-thigh and mid-shank regions and amputated, preserving the replaced knee with its soft tissue cover. The exposed shafts were potted using a combination of fine stone pebbles and dental stone (Golden Denstone Labstones, Modern Material, Kulzer LLC, South Bend, IN, USA). The prepared knee replacement specimen was then attached to a hydraulic powered 6-degree-of-freedom VIVO joint motion simulator (AMTI, Boston, MA, USA), using the same technique and alignment principles described in the previous literature [21,22] (Figure 6). The soft tissues were closed in a standard fashion with an ethibond suture. The simulator was set up in force control mode to manipulate the knee replacement through multiple cycles of different activities of daily living based on force and torque measurements imported from Orthoload, AVER75 dataset (www.orthoload.com) [23], except for flexion, which was displacement-controlled. In all cases, loads were reduced by 50% to avoid causing undue damage to the specimen. Similarly, for select activities (jogging, stair ascent, stair descent), the waveform was applied at half physiological speed (Table 1). The simulator’s iterative learning control (ILC) algorithm was employed, allowing the system to learn from its own tracking error and reduce said error over time [24]. Once we were satisfied with the smoothness of the simulator operation, we then proceeded to the next phase of the experiment.

We removed sutures from the knee to access the blind cavities in the prosthesis. We then placed identical IMU devices consisting of an accelerometer, gyroscope, and Bluetooth low-energy (BLE) module attached to a micro lithium polymer battery in the blind cavities (Figure 7). The IMU devices were sealed with heat shrink tubing for protection against body fluids and metal contact. The devices and battery fitted snugly in the cavities created in the prosthesis, ensuring they moved in unison with the prosthesis. The devices were programmed to continuously stream accelerometer and gyroscope data when connected to the live rechargeable battery.

A computer programme was created for the acquisition of the streamed IMU data with a graphic user interface to independently monitor and control the data input from the two devices (Figure 8). The powered devices were embedded within the prosthesis, the knee soft tissues were closed, and the VIVO simulator was activated for multiple repetitions (reps) of specific ADLs.

Activities simulated based on OrthoLoad in vivo measurements were

-Jogging (6 km/h on treadmill);-Knee bending;-Sitting down (from standing position);-Standing up (from seated position);-Walking (level ground);-Standing still;-Stair ascent;-Stair descent.

A minimum of 100 reps of each complete activity were collected per activity class. Jogging, walking, standing still, and stair navigation data was streamed and collected continuously. In activities that required starting the limb in a neutral position, like knee bending and sitting down, the data was collected from the neutral position (standing) to the end of the activity, excluding recovery back to the neutral position. The process was reversed for the standing up activity. The VIVO simulator was allowed to cycle through multiple reps of these activities, and data collection was activated and disabled manually based on visual inspection of the limb position.

## 3. Data Collection and Processing

IMU data was generated at 50 Hz and streamed in real time.

### 3.1. Synchronization of Data from Multiple IMU Devices

The dual IMUs did not activate in perfect synchrony and streamed data asynchronously. To correct for asynchronous activation and occasional missed data of the IMUs in the datasets, we binned the data into time-based windows of 100 ms bins to align the readings. For consistency, each bin retained only the most recent sample before the bin boundary. We generated sample windows of 30 readings for data analysis (Figure 9).

### 3.2. Data Preprocessing Including Cleanup and Filtering

Each IMU recorded six channels (three-axis accelerometer and three-axis gyroscope) at 50 Hz. Data from both IMUs were combined into a single 12-channel stream, synchronized into 100 ms bins, and windowed into 30-timestep samples (i.e., 3 s of activity per input window).

We applied a low-pass filter to the raw accelerometer data to remove high-frequency noise and modulate the effects of sudden movement and vibration. We applied a high-pass filter to remove drift and noise from the raw gyroscope data [25] (Figure 10).

We converted the raw data inputs of the three accelerometer and gyroscope readings (x, y, z) for a total of 12 spectrogram images. The spectrogram is effectively a time-frequency representation of the IMU signal, where the colour code represents the signal’s power (amplitude squared) at a given frequency over time. This transformation is useful because convolutional neural networks (CNNs) excel at learning spatial features in two-dimensional images and spectrograms convert time-series data into an image-like representation. By leveraging this approach, CNNs can capture frequency–time patterns that may be lost in purely time-domain analysis (Figure 11).

### 3.3. Artificial Intelligence Model Architecture

We implemented a dual-branch deep learning model combining a CNN and an LSTM (long-short term memory network) to classify activities of daily living from 12-channel IMU data. The dual-branch CNN–LSTM design leverages complementary strengths: the LSTM captures sequential dependencies in raw filtered time-series, while the CNN extracts discriminative spatiotemporal patterns from spectrograms. Combining these modalities yields improved robustness and accuracy compared to using either alone. The input data was windowed into 30-step samples and split into approximately 70% training (455 samples), 11% validation (75), and 19% testing (129).

The LSTM branch processes filtered time-domain sequences using two-layer LSTM (384 and 256 units) with dropout (20% and 45%).

The CNN branch processes spectrograms computed from the same input (shape: 8 × 8 × 12). It consists of a 2D convolutional layer (128 filters, 3 × 3), max pooling, dropout (40%), and a dense layer (256 units) before flattening.

The outputs of both branches are concatenated, passed through a fusion dense layer (256 units, ReLU), and fed into a final softmax layer predicting one of eight activity classes. The model was trained with categorical cross-entropy loss and the Adam optimizer (Figure 12).

## 4. Results

The final AI model achieved 95.68% out-of-sample test accuracy across eight distinct activity classes. Class-level evaluation revealed perfect classification (100% precision and recall) for static and transitional actions including knee bending, sitting down, stance, and standing up. More dynamic activities such as walking and stair navigation also performed strongly. Walking upstairs reached 96.8% recall, while walking downstairs—the most error-prone class—still achieved 87.5% recall and a 0.91 F1 score.

Overall, the macro-averaged F1 score was 0.96, indicating consistent generalization across classes, with strong robustness even in the presence of moderate class imbalance and temporal overlap between locomotion patterns (Table 2).

The confusion matrix represented below (Figure 13) summarizes classification performance across all 139 test samples (19% of the dataset as per Table 2). Each row shows true labels, with each column showing predicted labels. The diagonal elements indicate correct classifications, while off-diagonal values represent misclassifications. For example, there was occasional confusion between stair ascent and descent. This confirms high per-class fidelity across the full held-out test set.

## 5. Discussion

In this simulated cadaveric experimental study, we have been able to demonstrate that ADLs can be recognized based on data generated by IMU devices embedded in a knee replacement prosthesis. We have built an artificial intelligence-based model that recognizes patterns generated by the IMU devices. From the outset, it was important to have a reliable and accurate model. We achieved this through several unique design elements. First, we used two independently moving IMU devices and second, our AI powered activity recognition model incorporates spectrograms in pattern recognition. The two IMU devices presented us with a rich data source of 12 data points, which increases accuracy in correctly identifying the ADL. However, this is at the cost of increased complexity of the activity recognition model. In the first instance, we had to solve the asynchronous firing of the IMUs once activated. When powered, the Bluetooth module advertises a signal that is then recognized by the data collection programme on a computer. We were able to achieve this by creating moving windows. We chose to build spectrograms into our model, as this is a well-developed aspect of artificial intelligence image recognition. By generating spectrograms from the patterns created by the streamed data, we were able to take advantage of this powerful AI tool to increase the accuracy of activity recognition. Similar spectrogram-based CNN approaches have been successfully applied in domains such as power signal disturbance classification [26] and audio emotion recognition [27], highlighting the generalizability of spectrogram features for time-series classification tasks.

We generated both accelerometer and gyroscope data at 50 Hz. Higher frequencies of data generation would likely improve accuracy further. However, this would be at the expense of higher energy requirements and shorter battery life for a likely marginal benefit in accuracy. In the study, we specifically chose to stream the IMU data despite the higher energy consumption to enable immediate recognition of connection issues, dysfunction of the IMUs, and depleted batteries.

There is an increasing use of commercially available wearable IMU devices in assessing activity post-TKR [4]. There is also increasing recognition of the limitations of PROMs, and a combination of objective measures and PROMs may present the most comprehensive picture regarding the outcome of the TKR procedure [28]. In fact, the use of these devices is associated with improved early outcomes [29]. Early work in the field of wearable devices in TKR focused on the intensity of activity but with increased sophistication of the devices, parameters such as step count, and cadence can now be assessed and recorded [30,31].

Wearable devices in the current context are only useful when charged, turned on, and worn correctly on the body of the TKR patient. This presents challenges with tolerance, discomfort, and compliance, in general [13]. Embedding IMU devices within a TKR prosthesis is an elegant solution to deal with these issues. The closest we have come to achieving this ideal is Persona IQ TKR (Zimmerbiomet, Warsaw, IN, USA) [32]. This commercially available smart knee replacement consists of a single IMU device that is incorporated into a tibia prosthesis extension piece. IMU data derived from Persona IQ recipients has been used to plot ideal recovery milestones represented as recovery curves. These recovery curves, adjusted for age, make it easy to identify patients that are falling below the expected norms of recovery, allowing for early recognition and appropriate corrective action [33]. We believe that it is desirable to have an IMU device in both the femoral and tibial prosthesis, as these parts of the knee joint can move independently and in concert.

The inability to perform ADLs renders one disabled. The aim of the TKR is to free the patient of any physical disability related to the knee. The level of disability is expected to improve during recovery, implying an improvement in the ability to perform ADLs with restoration of the pre-disease functional state at the conclusion of rehabilitation. Supervised physical therapy is necessary to ensure that the patient is safe in the immediate post-operative period [34]. The ability to walk, sit, stand, and negotiate stairs are all important ADLs on the road to recovery of the patient. The ability to accurately recognize ADL activity post-TKR would be a major advancement in the current standard of quantification of knee kinematic activity. We believe that multiple IMU sensor devices present a much richer data stream than a single device and conversely improve the accuracy of ADL activity recognition. Continuous monitoring and seamless reporting of ADLs throughout rehabilitation would provide real-time feedback to healthcare practitioners while maintaining patient engagement and offering unprecedented levels of personalized rehabilitation [35]. To the best of our knowledge, an objective means of reliably assessing ADLs post-TKR remotely does not exist.

Limitations of the study include the use of a single cadaver knee and limited data generation per ADL. Additionally, the datasets used in creating the AI model were small. We also had power constraints in the limited battery life of the fully charged micro-battery. It may also be argued that using a simulator for repeated movements does not account for natural vagaries in movement. However, the simulator used is hydraulic actuated and is subject to vibrations unlike electric actuated simulators. Furthermore, the study is limited to a posterior stabilized implant which constrains movement patterns by design. Translating this proof-of-concept study to a smart knee replacement empowered to relay information on ADLs in a human is a difficult undertaking, owing to space constraints in a conventional TKR prosthesis. Accommodating the powered sensor devices while maintaining the structural integrity of the prosthesis is a challenge. Furthermore, the availability of batteries with sufficient long-term energy in a small form factor, vagaries of individual knee kinematics, sensor drift, cost implications, and regulatory hurdles are all factors that bear consideration.

## 6. Conclusions

Dual IMUs embedded in knee replacement prosthesis generated data that was analyzed using our proposed AI model, showing promising results with a high level of accuracy for activity recognition in a simulated cadaver model. Ultimately, we need larger studies using different designs of knee replacements to test the robustness of the model before embarking on human trials. This approach lays the foundation for fully integrated multi-sensor smart prosthetics capable of real-time functional monitoring and adaptive rehabilitation.

## 7. Patents

The subject matter of this manuscript has been registered as a provisional patent and is being filed as a full patent.

## Figures and Tables

**Figure 1 sensors-25-06657-f001:**
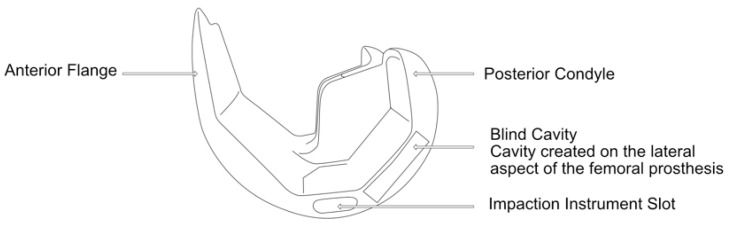
Illustration of lateral view of femoral prosthesis.

**Figure 2 sensors-25-06657-f002:**
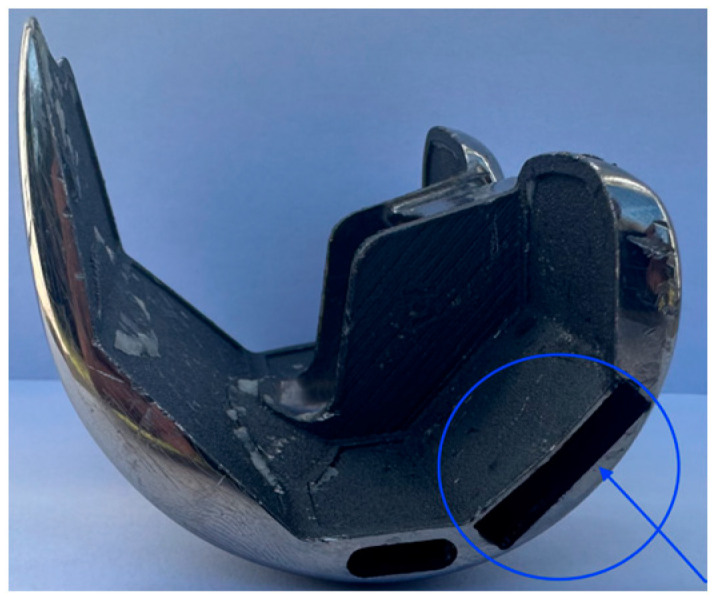
Lateral view of actual implant (blind chamber circled).

**Figure 3 sensors-25-06657-f003:**
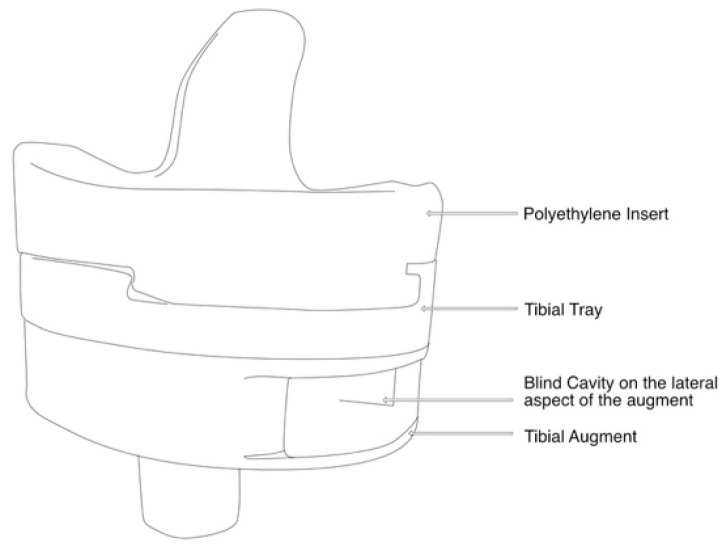
Illustration of lateral view of tibial prosthesis.

**Figure 4 sensors-25-06657-f004:**
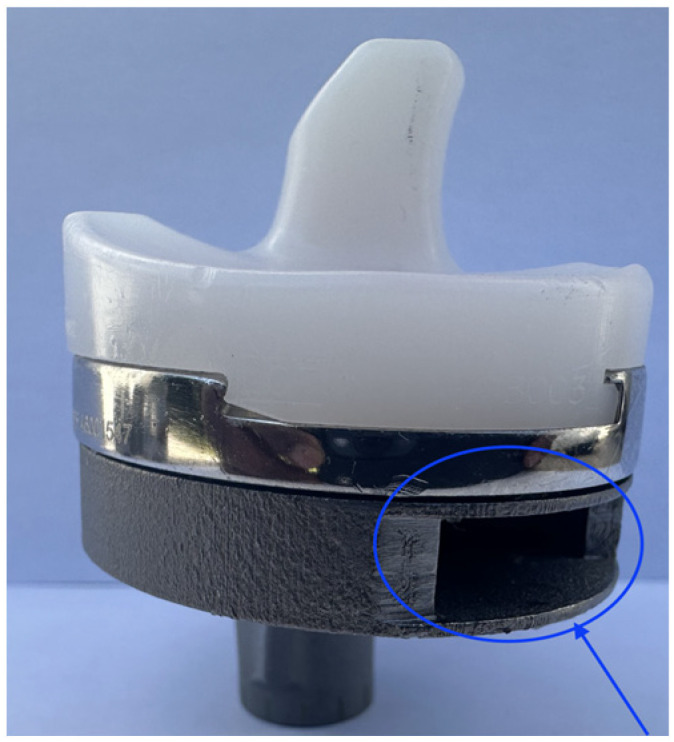
Lateral view of actual tibial prosthesis (blind chamber circled).

**Figure 5 sensors-25-06657-f005:**
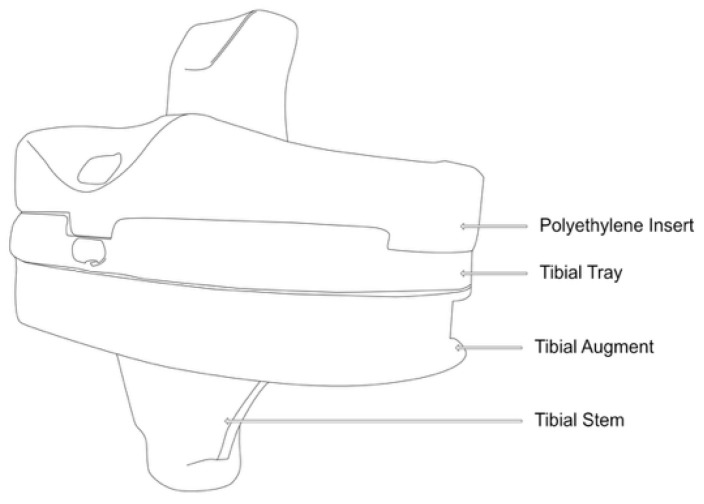
Illustration of oblique view of tibial prosthesis.

**Figure 6 sensors-25-06657-f006:**
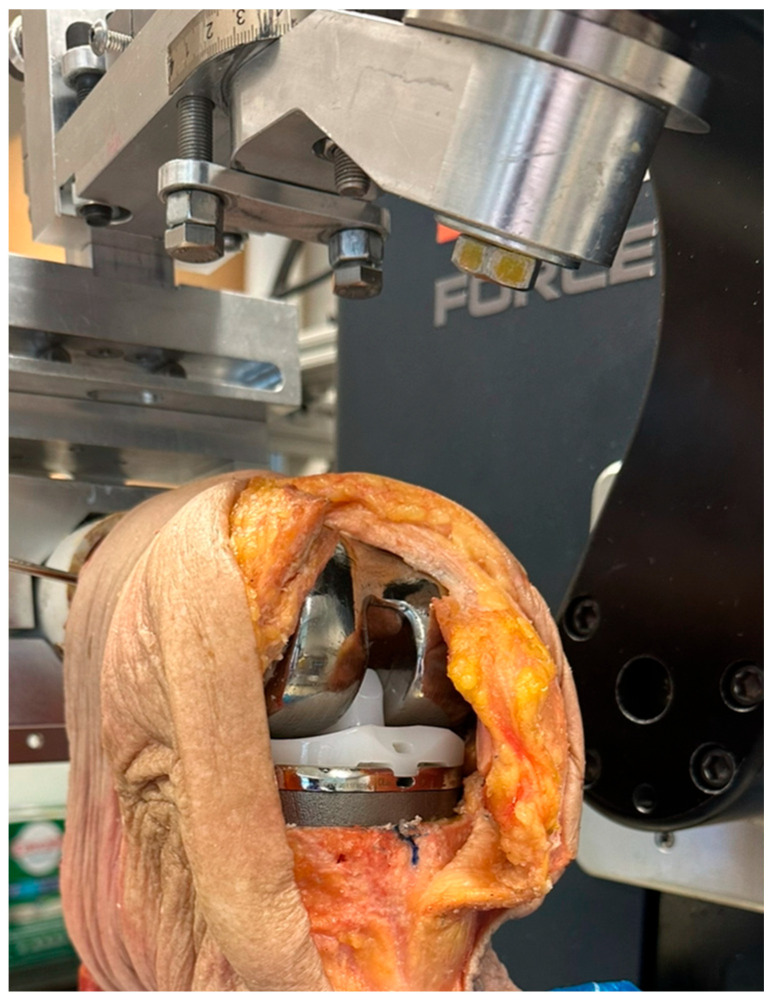
Cadaver knee with implanted TKR mounted on a simulator.

**Figure 7 sensors-25-06657-f007:**
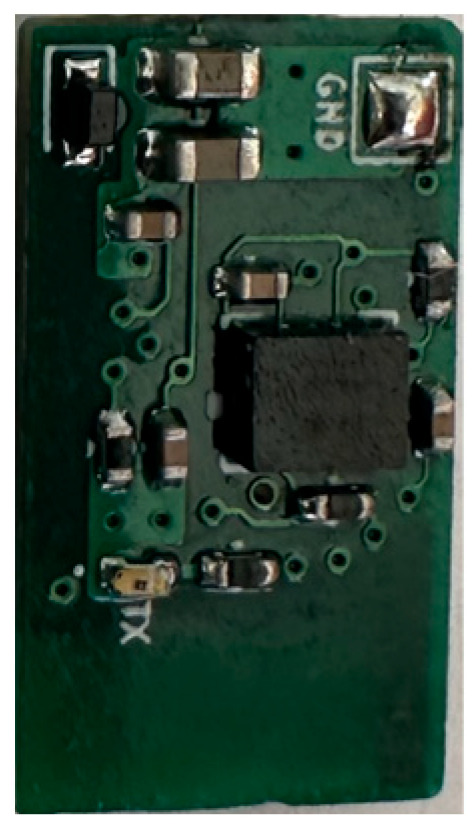
IMU device PCB programmed to sample 6 channels at 50 Hz.

**Figure 8 sensors-25-06657-f008:**
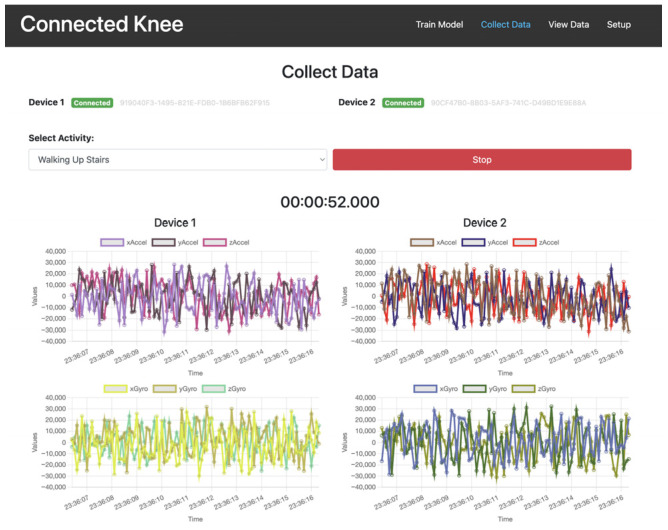
Graphic user interface of multi-IMU device data collection tool.

**Figure 9 sensors-25-06657-f009:**
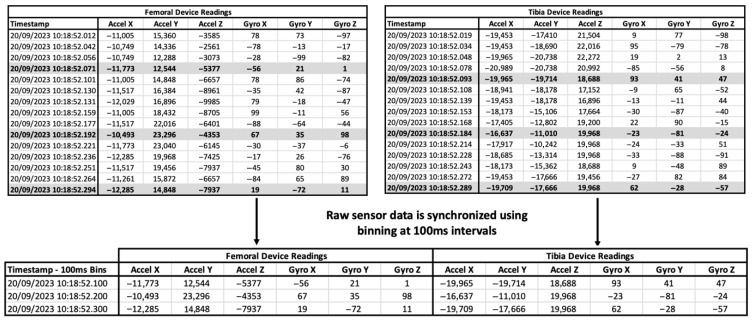
Example of raw data synchronization for dual IMUs. The binned data from the data streams is highlighted in grey and captured in the bottom table.

**Figure 10 sensors-25-06657-f010:**
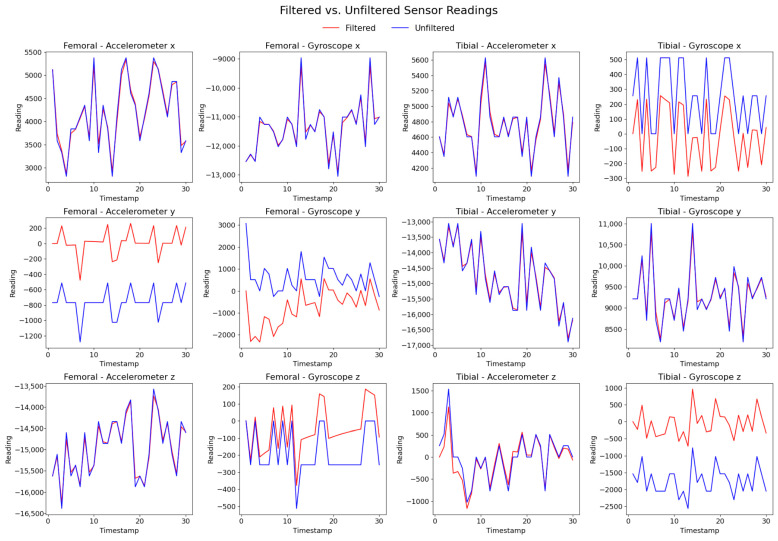
Effect of filtering the raw data.

**Figure 11 sensors-25-06657-f011:**
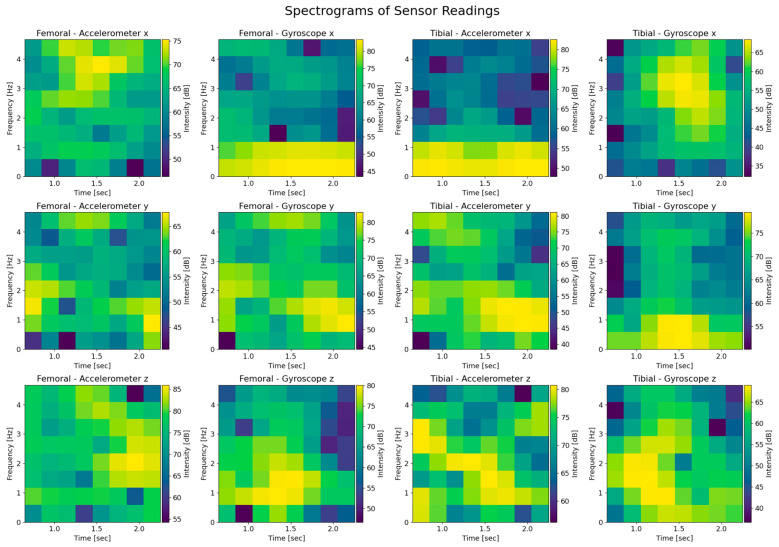
Spectrograms of the raw data.

**Figure 12 sensors-25-06657-f012:**
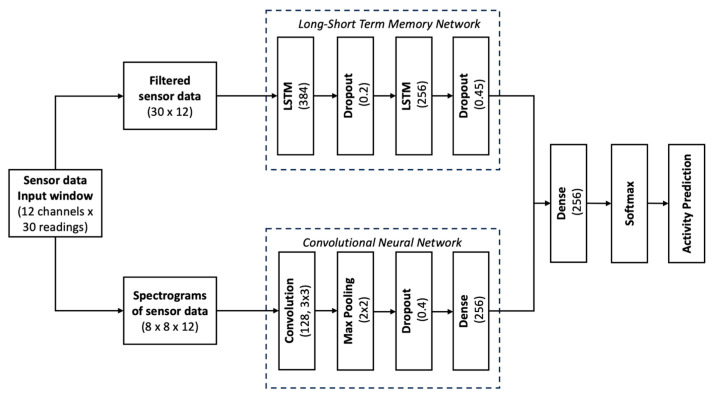
Diagram showing AI model architecture.

**Figure 13 sensors-25-06657-f013:**
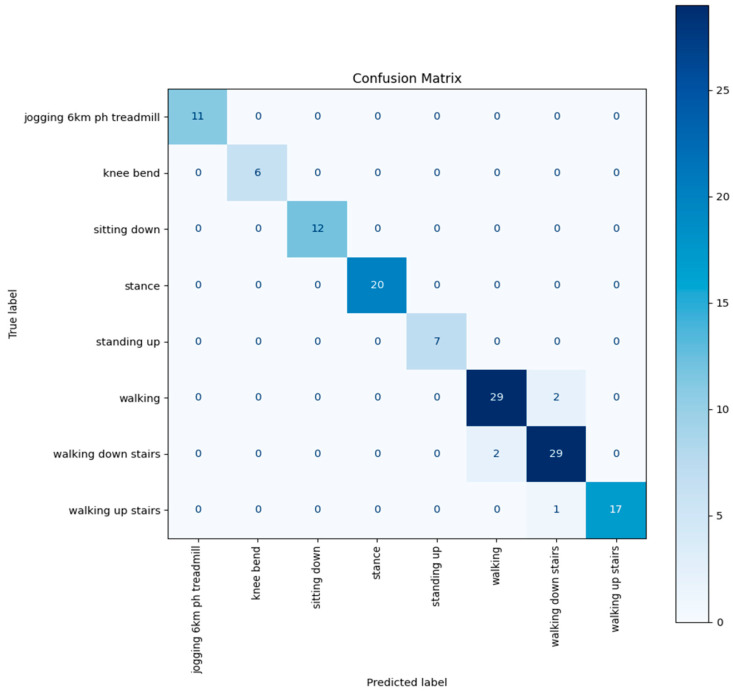
AI model confusion matrix.

**Table 1 sensors-25-06657-t001:** Flexion range and motion and waveform for simulated knee movements.

Motion Simulated Based on Orthoload	Flexion Range of Motion [Degrees]	Tested Rate [Hz] (Original Rate)
Walking (level ground)	7–62	0.94 (0.47)
Jogging (6 km/h on treadmill)	7–62	1.4 (0.7)
Stair ascent	13–94	0.56 (0.28)
Stair descent	14–98	0.58 (0.29)
Sitting down (from standing position)	6–94	0.16 (0.16)
Standing up (from seated position)	6–94	0.16 (0.16)
Knee bending	4–98	0.13 (0.13)
Standing still	7–13	0.1 (0.1)

**Table 2 sensors-25-06657-t002:** Model results and analysis.

Activity	Precision	Recall	F1 Score	Support
Jogging 6 km/h treadmill	1.00	0.80	0.89	10
Knee bending	1.00	1.00	1.00	3
Sitting down	1.00	1.00	1.00	10
Stance	1.00	1.00	1.00	22
Standing up	1.00	1.00	1.00	10
Walking	0.97	1.00	0.98	31
Walking downstairs	0.95	0.88	0.91	24
Walking upstairs	0.88	0.97	0.92	31
Accuracy (overall)	—	—	0.96	139
Weighted average	0.96	0.96	0.96	139

Dash in the table represents not applicable.

## Data Availability

The data presented in this study is available upon request from the corresponding author due to commercial restrictions.

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
