# Peer review of "Smart Total Knee Replacement: Recognition of Activities of Daily Living Using Embedded IMU Sensors and a Novel AI Model in a Cadaveric Proof-of-Concept Study†"

_sensors, 2025, doi:10.3390/s25216657_

Round 1

Reviewer 1 Report

Comments and Suggestions for Authors

This paper discusses the use of IMU devices embedded in a prosthesis for total knee replacement (TKR). In this “proof-of-concept” study, the authors demonstrate that such devices allow the recognition of activities of daily living (ADL), when executed by a knee simulator and evaluated by a dual-branched deep learning model consisting of CNN and LSTM networks.

General:

Overall, this is an interesting and well-written paper, with a project idea that is very early in its conception. Tagging it as a “proof-of-concept” study would classify it correctly. The introduction could be expanded with a short review of ADL assessments for TKR subjects. In addition, some background about the dual branched neural network and its use should be added. The Discussion misses a through and honest debate about the problems that will be faced by translating the idea to in vivo conditions, for example,

(1) how will the IMUs be powered and are those batteries safe and how long will they last?  
(2) How will be drift of the IMU sensors be handled?
(3) Is the prosthesis still safe and won’t fracture after IMU chamber machining?
(4) Can you still expect accuracy of the AI network when actual TKR subjects with several (abnormal) gait patterns walk with it?
(5) What about costs? Will there be a net benefit of such a device and why?

None of the Figures and Tables are cited in the text! Please generate references! In many of the presented figures the legend on axes is illegible

Specific:

Title: Please add “a proof-of-concept study” to the title (you could do this by deleting ‘smart total knee replacement’)

Abstract, line 28” No need to use ‘propriety powered’ here.

Introduction, Line 52ff: specific background of activity recognition in TKR should be given (e.g. PMID: 38359050, PMID: 38823517, PMID: 26347875, PMID: 40648199, PMID: 18524001, PMID: 38428687)

Line 57: Reference 6 is outdated to provide current TKR numbers.

Methods, Line 121: Orthoload provides datasets of several individuals. Which one was picked? Or were all the data used and averaged?

Line 129: Why were the IMUs inserted after the TKR was implanted? This would not be possible during a real surgery.

Lines 165-166: The presentation of Figure 6 is not necessary. Delete!

Line 184-185: Explanation and background on the spectrogram is missing in the text.

Line 191-192: Please provide the full name of the abbreviations CNN and LSTM and describe and comment on the specific advantages of the network structure. (There are some lines about it in the discussion (lines 236-39). This needs to be provided earlier.)

Line 206: Please remove this sentence, unless you made the dataset available to my co-reviewer(s). [In that case, please mention in your rebuttal]

Line 222: Please provide text and explanation about the use of the confusion matrix (Figure 11)

Discussion, Lines 287-288: The reviewer would disagree that hydraulics is a source of waveform variation.

Please note the points noted under “general”

Figure 1: A technical drawing showing the chambers would be helpful.

Figure 6: Delete!

Figure 8: Can’t read axes and legends

Figure 9: Can’t read axes. What is shown in color code? Can you add explaining text to the caption?

Figure 11: Legend and matrix entry are too small to read. Also, if you could add text about its meaning to the caption!

Author Response

Reviewer 1

This paper discusses the use of IMU devices embedded in a prosthesis for total knee replacement (TKR). In this “proof-of-concept” study, the authors demonstrate that such devices allow the recognition of activities of daily living (ADL), when executed by a knee simulator and evaluated by a dual-branched deep learning model consisting of CNN and LSTM networks.

General:

Overall, this is an interesting and well-written paper, with a project idea that is very early in its conception. Tagging it as a “proof-of-concept” study would classify it correctly. The introduction could be expanded with a short review of ADL assessments for TKR subjects. In addition, some background about the dual branched neural network and its use should be added. The Discussion misses a through and honest debate about the problems that will be faced by translating the idea to in vivo conditions, for example,

(1) how will the IMUs be powered and are those batteries safe and how long will they last?  
(2) How will be drift of the IMU sensors be handled?
(3) Is the prosthesis still safe and won’t fracture after IMU chamber machining?
(4) Can you still expect accuracy of the AI network when actual TKR subjects with several (abnormal) gait patterns walk with it?
(5) What about costs? Will there be a net benefit of such a device and why?

Thank you for pointing this out.  We agree with this comment.  Therefore, we have added the following sentence in the discussion in response to the above:

Line 331

Translating this proof-of-concept study to a smart knee replacement empowered to relay information on ADLs in a human is a difficult undertaking owing to space constraints in conventional  TKR prosthesis.  Accommodating the powered sensor devices while maintaining structural integrity of the prosthesis is a challenge.  Furthermore, availability of batteries with sufficient long-term energy in a small form factor, vagaries of individual knee kinematics, sensor drift, cost implications and  regulatory hurdles are all factors that bear consideration.

None of the Figures and Tables are cited in the text! Please generate references! In many of the presented figures the legend on axes is illegible

Thank you for pointing this out.  We agree with this comment.  Therefore, we have cited all the figures and tables in the text.  We have increased the font on the axes to improve legibility

Line 123,152,194,205,218,240,253,256,257.

Line 207, 221

Specific:

Title: Please add “a proof-of-concept study” to the title (you could do this by deleting ‘smart total knee replacement’)

Thank you for pointing this out.  We agree with this comment.  Therefore, we have changed the title and added proof-of-concept study

Line 2

Smart Total Knee Replacement: Recognition of Activities of Daily Living Using Embedded IMU Sensors and a Novel AI Model in a Cadaveric Proof-of-Concept Study

Abstract, line 28” No need to use ‘propriety powered’ here.

Thank you for pointing this out.  We agree with this comment.  Therefore, we have removed the word proprietary from the text

Introduction, Line 52ff: specific background of activity recognition in TKR should be given (e.g. PMID: 38359050, PMID: 38823517, PMID: 26347875, PMID: 40648199, PMID: 18524001, PMID: 38428687)

Thank you for pointing this out.  We agree with this comment.  Therefore, we have expanded the section to highlight the developments in IMUs.  We have reworked the introduction to accommodate this section.  We have added five of the most relevant references.

Line 70,71,73,74,78 (references)

Line 69-78

Wearable devices have been used to establish normative values for daily functional recovery patterns following TKR based on total daily steps and cadence(13).  Wearables have also demonstrated accurate knee angle measurements post TKR across ADLs(14).  A recent study on an implanted IMU housed in a tibial extension of a knee replacement prosthesis showed objective increases in gait parameters in the first six weeks(15).  This IMU-generated activity data from implanted devices has been used to create recovery curves based on quantitative metrics like step counts (16).   Published TKR research involving use of IMUs has focused on spatiotemporal metrics largely excluding the wider spectrum of ADLs(17).  Specific ADLs studied using a wearable activity monitor following TKR showed the most frequently performed activity was standing, followed by level walking, sitting, stair walking, and lying down(18).

Line 57: Reference 6 is outdated to provide current TKR numbers.

Thank you for pointing this out.  We agree with this comment.  We fully agree with your assertion.  We wanted to present a more global perspective for the number of knee replacements performed as opposed to simply quoting numbers from the US, UK, Scandinavia or other limited jurisdiction hence, the chosen reference.  We have failed to identify a single publication with as wide a global perspective as the cited paper.  We are astonished that even leading publications on global disease burdens (Price, Andrew J, Abtin Alvand, Anders Troelsen, Jeffrey N. Katz, Gary Hooper, Alastair Gray, Andrew Carr, and David Beard. "Knee replacement." The Lancet 392, no. 10158 (2018): 1672-1682) choose to limit their estimates of global TKR to select countries. 

Methods, Line 121: Orthoload provides datasets of several individuals. Which one was picked? Or were all the data used and averaged?

Line 137

We used the AVER75 (average loads in subjects with 75kg body weight dataset.  We have made reference to AVER75 dataset in the text.

Line 129: Why were the IMUs inserted after the TKR was implanted? This would not be possible during a real surgery.

Thank you for pointing this out.  The rechargeable batteries could only power the IMU for a limited time period.  The IMUs were fired and confirmed to be transmitting data before they were implanted into the knee prosthesis.  The knee was then sutured closed, the simulator was started and data was subsequently collected.  Fresh batteries were swapped for dead batteries as was necessary.  The positioning of blind cavities to house the IMU and battery on the lateral aspect of the prosthesis is a huge advantage over a sensor device in the tibial extension of the tibia prosthesis as in the current smart knee replacement on the market.  The lateral aspect of the prosthesis is relatively accessible even at the end the procedure should there be malfunction or dysfunction of the sensor. We believe that it would be entirely possible to insert the IMUs after implantation of the knee replacement in this instance. 

Lines 165-166: The presentation of Figure 6 is not necessary. Delete!

Thank you for pointing this out.  We agree with this comment.  Therefore, we have deleted fig 6

Line 184-185: Explanation and background on the spectrogram is missing in the text.

Thank you for pointing this out.  We agree with this comment.  Therefore, we have added the following text to explain the spectrograms. 

Line 214-220

We converted the raw data inputs of the three accelerometer and gyroscope readings (x,y,z)  for a total 12 image spectrogram per channel.  The spectrogram is effectively a time-frequency representation of the IMU signal, where the colour code represents the signal’s power (amplitude squared) at a given frequency over time.  This transformation is useful because convolutional neural networks (CNN) excel at learning spatial features in 2-Dimensional images  and, spectrograms convert time-series data into an image-like representation. By leveraging this approach, CNNs can capture frequency-time patterns that may be lost in purely time-domain analysis.   

We have included additional details on the basis for including the spectrogram in the model architecture in the discussion section. 

Line 283-285

Similar spectrogram-based CNN approaches have been successfully applied in domains such as power signal disturbance classification and audio emotion recognition, highlighting the generalizability of spectrogram features for time-series classification tasks.

Line 191-192: Please provide the full name of the abbreviations CNN and LSTM and describe and comment on the specific advantages of the network structure. (There are some lines about it in the discussion (lines 236-39). This needs to be provided earlier.)

Thank you for pointing this out.  We agree with this comment.  Therefore, the full names have been added and the following text has been inserted on the specific advantages.

Line 217, 228 (full names)

Line 229

The dual-branch CNN–LSTM design leverages complementary strengths: the LSTM captures sequential dependencies in raw filtered time-series, while the CNN extracts discriminative spatiotemporal patterns from spectrograms. Combining these modalities yields improved robustness and accuracy compared to using either alone. 

Line 206: Please remove this sentence, unless you made the dataset available to my co-reviewer(s). [In that case, please mention in your rebuttal]

Thank you for pointing this out.  We agree with this comment.  Therefore, the sentence has been removed.

Line 222: Please provide text and explanation about the use of the confusion matrix (Figure 11)

Thank you for pointing this out.  We agree with this comment.  Therefore, we have added the text below to explain the confusion matrix

Line 258 – 262

The confusion matrix represented below (fig 11) summarizes classification performance across all 139 test samples (19% of the dataset as per Table 2).  Each row shows true labels with each column showing predicted labels.  The diagonal elements indicate correct classifications, while off-diagonal values represent mis-classifications.  For example, occasional confusion between stair ascent and descent.  This confirms high per-class fidelity across the full held-out test set.

Discussion, Lines 287-288: The reviewer would disagree that hydraulics is a source of waveform variation.

Thank you for pointing this out.  We do not agree with this comment.
The statement is made in comparison to electric actuated simulators.  The hydraulic actuated machine operates in force control and exact replication of joint kinematics from one cycle to the next is not guaranteed. For instance, frictional characteristics change in the long-term, and random vibrations occur in the short-term. Furthermore, the iterative learning controller (ILC) means that the simulator is constantly working to reduce force errors, and this causes kinematics to evolve over time.

Please note the points noted under “general”

Figure 1: A technical drawing showing the chambers would be helpful.

Thank you for pointing this out.  We agree with this comment.  Therefore, a technical drawing has been added to the pictures

Line 106,109,112

Figure 6: Delete!

Deleted

Figure 8: Can’t read axes and legends

Thank you for pointing this out.  We agree with this comment.  Therefore, axes and legends have been enlarged and are now legible

Figure 9: Can’t read axes. What is shown in color code? Can you add explaining text to the caption?

Thank you for pointing this out.  We agree with this comment.  Therefore, axes are now legible – increased font 30%.  Text below has been added. 

Line 214-220

We converted the raw data inputs of the three accelerometer and gyroscope readings (x,y,z)  for a total 12 spectrogram images.  The spectrogram is effectively a time-frequency representation of the IMU signal, where the colour code represents the signal’s power (amplitude squared) at a given frequency over time.  This transformation is useful because convolutional neural networks (CNN) excel at learning spatial features in 2-Dimensional images  and, spectrograms convert time-series data into an image-like representation.  By leveraging this approach, CNNs can capture frequency-time patterns that may be lost in purely time-domain analysis. 

Figure 11: Legend and matrix entry are too small to read. Also, if you could add text about its meaning to the caption!

Thank you for pointing this out.  We agree with this comment.  Therefore, the picture has been enlarged and the text below has been added.

Line 258-262

The confusion matrix represented below (fig 13) summarizes classification performance across all 139 test samples (19% of the dataset as per Table 2).  Each row shows true labels with each column showing predicted labels.  The diagonal elements indicate correct classifications, while off-diagonal values represent mis-classifications.  For example, occasional confusion between stair ascent and descent.  This confirms high per-class fidelity across the full held-out test set.

Reviewer 2 Report

Comments and Suggestions for Authors

This paper describes the development of an IMU sensing system that is embedded within a total knee replacement and uses an AI model to classify knee activity in a cadaveric model. Two IMU devices featuring a Bluetooth Low Energy (BLE) module and a battery were embedded within the femoral and tibial components of the total knee replacement prosthesis that was then implanted into a cadaveric leg. The cadaveric leg was then loaded into a joint simulator and manipulated the leg through 8 motions simulating 8 activities of daily living whilst data was recorded by the IMUs. An AI model was then trained to classify the data collected from the IMUs into the 8 activities. The results showed a high level of activity recognition and the authors concluded that this technology could one day be included within a smart prosthesis.

  1. Introduction:

Introduction does a good job of outlining clinical need, current practices and associated technology.

Ln 44: “However, these facilities are expensive to set up and access, require specialized expertise, and are created in controlled environments that may not reflect real-world conditions.” – please include references for this statement

  1. Materials and Methods:

Ln 94 please provide more specifics on the ‘rapid prototyping technology’

Ln 161 Because you manually activated and disabled data collection were there instances where data collection was activated whilst the simulator was stationary? If so what impact would this have on the model performance and did the visual & manual inspection introduce significant time discrepancies?

Figure 6 not sure this adds a great deal.

  1. Data Collection and Processing:

Ln 163 unsure what this figure of raw data is adding and if it is necessary.

  1. Synchronization of data from multiple IMU devices:

Ln 168 can you provide more clarification on the how you binned the data and how this synchronised the data and how Figure 7 explains the method used.

  1. Data preprocessing including cleanup and filtering

Ln 184 please explain Figure 9 and provide some context to the plots.

  1. Artificial Intelligence Model Architecture

Ln 191 introduce acronyms in main body of text CNN and LSTM.

  1. Results

Table 2. and Figure 11. Please introduce theses in the main body of text and supply some more description of what the confusion matrix shows. Does this represent all the data? The you state that 19% testing (129) was used for testing but 139 was the total shown in table 2.

  1. Discussion

Good discussion which provided context to the research and where it sits in the field and weighed up advantages of smart prosthesis compared to wearable devices.

Overall I would recommend publication taking into account the comments above.

Author Response

Reviewer 2

Submission Date

27 July 2025

Date of this review

05 Aug 2025 18:12:29

This paper describes the development of an IMU sensing system that is embedded within a total knee replacement and uses an AI model to classify knee activity in a cadaveric model. Two IMU devices featuring a Bluetooth Low Energy (BLE) module and a battery were embedded within the femoral and tibial components of the total knee replacement prosthesis that was then implanted into a cadaveric leg. The cadaveric leg was then loaded into a joint simulator and manipulated the leg through 8 motions simulating 8 activities of daily living whilst data was recorded by the IMUs. An AI model was then trained to classify the data collected from the IMUs into the 8 activities. The results showed a high level of activity recognition and the authors concluded that this technology could one day be included within a smart prosthesis.

  1. Introduction:

Introduction does a good job of outlining clinical need, current practices and associated technology.

Ln 44: “However, these facilities are expensive to set up and access, require specialized expertise, and are created in controlled environments that may not reflect real-world conditions.” – please include references for this statement

Thank you for pointing this out.  We agree with this comment.  Therefore, we have added the following two references.

Line 46

 Bini SA, Gillian N, Peterson TA, Souza RB, Schultz B, Mormul W, et al. Unlocking Gait Analysis Beyond the Gait Lab: High-Fidelity Replication of Knee Kinematics Using Inertial Motion Units and a Convolutional Neural Network. Arthroplast Today. 2025 Jun 1;33.

  1. Motoi, S. Tanaka, Y. Kuwae, T. Yuji, Y. Higashi, T. Fujimoto, and K. Yamakoshi, “Evaluation of a Wearable Sensor System Monitoring Posture Changes and Activities for Use in Rehabilitation,” J. Robot. Mechatron., Vol.19 No.6, pp. 656-666, 2007.

  1. Materials and Methods:

Ln 94 please provide more specifics on the ‘rapid prototyping technology’

Thank you for pointing this out.  We agree with this comment.  Therefore, we added the following text to expand on the rapid prototyping technology.

Line 102-105

The augment was created from 3-dimensional CAD drawings and manufactured using laser sintering technology for additive manufacturing on a EOSINT M280 3D printer (Centre for Rapid Prototyping and Manufacture, Bloemfontein, South Africa) and affixed securely to the underside of a size 7 tibial component using screws. 

Ln 161 Because you manually activated and disabled data collection were there instances where data collection was activated whilst the simulator was stationary? If so what impact would this have on the model performance and did the visual & manual inspection introduce significant time discrepancies?

Thank you for pointing this out.  The IMUs were activated while the simulator was stationary.  The simulator was then activated for specific ADL and data was only collected using the connected knee data collection tool after multiple repetitions of the activity.  The timing of the activation of the data collection was based on visual inspection.  We do not believe that the visual inspection trigger introduced significant time discrepancies as more than a 100 cycles were collected for each ADL class and possible discrepancies would have been smoothened by our data synchronization techniques.

Figure 6 not sure this adds a great deal.

The figure has been removed

  1. Data Collection and Processing:

Ln 163 unsure what this figure of raw data is adding and if it is necessary.

 The figure has been removed

  1. Synchronization of data from multiple IMU devices:

Ln 168 can you provide more clarification on the how you binned the data and how this synchronised the data and how Figure 7 explains the method used.

Thank you for pointing this out.  We agree with this comment.  Therefore, we re-worked the sentence to provide more clarification on the binning process. 

Line 192-196

 The dual IMUs did not activate in perfect synchrony and streamed data asynchronously.  To correct for asynchronous activation and occasional missed data of the IMUs in the datasets, we binned the data into time-based windows 100ms bins to align the readings.  For consistency, each bin retained only the most recent sample before the bin boundary.  We generated sample windows of 30 readings for data analysis.

  1. Data preprocessing including cleanup and filtering

Ln 184 please explain Figure 9 and provide some context to the plots.

Thank you for pointing this out.  We agree with this comment.  Therefore, we have added the following text

Line 214-220

We converted the raw data inputs of the three accelerometer and gyroscope readings (x,y,z)  for a total of 12 spectrogram images.  The spectrogram is effectively a time-frequency representation of the IMU signal, where the colour code represents the signal’s power (amplitude squared) at a given frequency over time.  This transformation is useful because convolutional neural networks (CNN) excel at learning spatial features in 2-Dimensional images  and, spectrograms convert time-series data into an image-like representation.  By leveraging this approach, CNNs can capture frequency-time patterns that may be lost in purely time-domain analysis (fig 11). 

  1. Artificial Intelligence Model Architecture

Ln 191 introduce acronyms in main body of text CNN and LSTM.

Thank you for pointing this out.  We agree with this comment.  Therefore, we have introduced the acronyms.

Line 217, 228

  1. Results

Table 2. and Figure 11. Please introduce theses in the main body of text and supply some more description of what the confusion matrix shows. Does this represent all the data? The you state that 19% testing (129) was used for testing but 139 was the total shown in table 2.

Thank you for pointing this out.  We agree with this comment.  Therefore, the picture has been enlarged and the text below has been added.

Line 258-262

The confusion matrix represented below (fig 13) summarizes classification performance across all 139 test samples (19% of the dataset as per Table 2).  Each row shows true labels with each column showing predicted labels.  The diagonal elements indicate correct classifications, while off-diagonal values represent mis-classifications.  For example, occasional confusion between stair ascent and descent.  This confirms high per-class fidelity across the full held-out test set.

  1. Discussion

Good discussion which provided context to the research and where it sits in the field and weighed up advantages of smart prosthesis compared to wearable devices.

Overall I would recommend publication taking into account the comments above.

Reviewer 3 Report

Comments and Suggestions for Authors

The reviewer was looking for evidence of verification of the technology and its application by the authors towards the problem.  This aspect seems to be lacking in the report.  Whilst the authors have mentioned the work of Bergmann and Orthoload.com the paper, work by Denis and Komistek have published detailed fluroscopic activity data for patients with joint replacements that would have been good to verify the current technology and simulator against.  Whilst the paper offers significant results the underlying calculations for analysis such as the test accuracy lacked specific detail leaving the reader having to accept the graphs without being able to firstly consider the underlying science that was engaged. Thus the methods and related discussion aspects to the methodology that was employed seems to lack rigour.  

Ultimately the reviewer feels that advantage of the device over a standard wearable activity monitor has not been sufficiently technically addressed by the authors to support the devices usage so an opportunity is lost that could be addressed with further scientific rigour.

Comments on the Quality of English Language

The report is written well, however, it should be written in 3rd person past tense to follow the recognised standard. 

Author Response

Reviewer 3

Thank you for pointing this out.  We agree with this comment. Comparing fluoroscopic data to IMU data would have been useful to validate our approach to activity recognition.  However, this would entail further methodological complexity at substantial increase in cost and beyond the scope of the current study.  Our aim was not to compare the embedded IMU devices to wearables.  However, it is common cause that wearables suffer from errors in recording sagittal knee angles particularly at extremes of movement inherent in activities like deep knee bend, sitting and standing up because of overlying tissue and skin movement.  We believe that IMUs embedded within the prosthesis would more accurately reflect the movement of the replaced knee compared to a body worn wearable. 

Round 2

Reviewer 1 Report

Comments and Suggestions for Authors

The authors addressed my concerns and I have no further questions/comments. Thanks for a nice rebuttal.